# The Pillars of Skill-Acquisition and Generalization; Why efficient General Intelligence requires Multi-Component Integration

## Abstract

Breakthroughs of Large Language Models (LLMs) have rekindled hopes for broadly capable artificial intelligence (i.e., Artificial General Intelligence (AGI)). Yet, these models still exhibit notable limitations – particularly in deductive reasoning and *efficient* skill acquisition. In contrast, *neuro-symbolic* approaches can exhibit more robust generalization across diverse tasks, as they integrate sub-symbolic pattern extraction with explicit logical structures. In this position paper, we go *a step further* and dissect generalizing systems into *six* pillars: well-defined model specificity, (human) capability encoding, dynamic knowledge acquisition & transfer, meaningful representations, abstraction & hierarchies, as well as the synergy effects resulting from component interactions. Based on historical and contemporary Artificial Intelligence (AI) approaches, we conclude that such *a multi-component implementation strategy is necessary for efficient general intelligence*. Our position is reinforced by the latest performance gains on the Abstraction and Reasoning Corpus (ARC) generalization benchmark.

## 1 Introduction

Across decades of progress, research on "artificial intelligence" has often centered on narrow tasks and small leaps in computational automation, without necessarily pursuing robust, human-like intelligence. This changed with the rise of large neural networks – models that excel in pattern extraction and display intriguing emergent capacities [Bubeck et al., 2023]. Yet, while these black-box approaches are remarkable in many respects, they also suffer from opaque decision-making processes and often exhibit only *local* forms of generalization. They thus provide limited insights into the core mechanisms underlying *flexible, human-level* intelligence.

Motivated by these gaps, an increasing number of researchers suggest incorporating symbolic reasoning into deep learning pipelines, giving rise to *neuro-symbolic* approaches [d'Avila Garcez and Lamb, 2023, Keber et al., 2024]. By preserving the neural model's strengths in statistical pattern recognition and combining them with symbolic structures that allow for compositional logic, explainable decisions, and interpretability, neuro-symbolic methods promise broader skill-acquisition efficiency, deeper semantic understanding, and safer real-world deployment [Hernández-Orallo, 2020, Hassija et al., 2024].

However, merely layering symbolic modules on top of neural back-ends does not automatically confer *general* intelligence. To foster meaningful progress, we must first **define** (1) the gist of (artificial) intelligence, especially in terms of *skill acquisition efficiency*, then pinpoint how best to **evaluate** (1) a model's capacity to abstract knowledge from sparse data and adapt to novel tasks. On this basis, we evaluate predominant research directions, such as Large Reasoning Models (LRMs) and neuro-

symbolic approaches (2). We derive our position that **a novel, multi-component implementation strategy is necessary for efficient general intelligence**. Our core contribution is the identification of six different fundamental pillars for achieving efficient generalization (3); (3.1) Model Specificity, (3.2) (Human) Capability Encoding, (3.3) Meaningful Representations, (3.5) Knowledge Acquisition & Transfer, (3.4) Abstractions & Hierarchies, and (3.6) Multi-Component Synergy. We conclude with implications for future research directions and practical system design (4).

**Defining Intelligence as Skill-Acquisition Efficiency**   Despite centuries of study, intelligence remains notoriously difficult to define comprehensively [Legg and Hutter, 2007]. We adopt the formulation by Chollet [2019] that views the intelligence of a system as "*a measure of its skill-acquisition efficiency over a scope of tasks, with respect to priors, experience, and generalization difficulty.*" This perspective shifts attention from raw *performance* on a single task to the *ability to learn new* tasks under constraints – such as limited data, novel transformations, or minimal prior knowledge.

An agent that extracts greater competence (skills, insights, etc.) from identical training conditions is inherently more efficient at acquiring skills. In evolution, this efficiency holds an inherent advantage. Most modern intelligence tests are based on the related concepts of fluid and crystallized intelligence. Figure 3 from [Chollet, 2019] visualizes this *information conversion ratio* from situational to operational space nicely. All other factors (e.g., prior knowledge, curriculum size, development efforts (e.g., inductive bias), training time & strategy, intrinsic task difficulty) should be *controlled for* to isolate the system's skill-acquisition efficiency. Even the competence a developer (or the development process) (un)consciously puts into the model (e.g., by hyperparameter choice) shall be accounted for to get a clean measure of the system's own skill-acquisition efficiency. For more on "developer-aware" generalization see Appendix B.

Of course, this intelligence definition has shortcomings but we believe[1] that it is currently the most *complete* and *correct* notion of intelligence we have, which is applicable to humans as well as machines. A consequence of this perspective is that **skill-acquisition efficiency** is at the heart of what sets "general" intelligence apart from specialized or over-engineered solutions [Bober-Irizar and Banerjee, 2024]. Hence, if the field's ambition is true *general* intelligence – rather than a proliferation of specialized or heavily handcrafted solutions – then adopting metrics and methods highlighting *skill-acquisition efficiency* becomes indispensable. This, in turn, requires reliable ways to evaluate how well a model performs under low-data, unseen, or compositional scenarios – where brute-force training or naive memorization is infeasible.

**Benchmarking for Generality**   Skill-acquisition efficiency, i.e., the amount of competence gained from a fixed amount of data or experience, should be evaluated independently of a system's final performance. The prior knowledge of a system matters as it gives a head start on performance. Traditional benchmarks often conflate performance with the data or developer-engineered interventions needed to achieve it. Here is where the ARC comes in; it tries to isolate and measure a system's inherent learning efficiency [Chollet, 2019]. The ARC benchmarks (currently in version ARC-AGI-1 and ARC-AGI-2) consist of small, diverse puzzles that test "core knowledge" concepts like spatial manipulation, color/object transformations, or compositional logic [Moskvichev et al., 2023]. Besides their simple format/setting, the crucial challenge is that no task is like the other, and the test set contains tasks that are unseen during training. The point of ARC is to force the model to learn the least common denominator of all scenarios, meaning that it needs to generalize over and flexibly operate on the underlying geometric concepts. This requires some form of generalization capability over a set of problems, making the benchmark robust against performance saturation by large, curated datasets.

Despite being straightforward for humans, ARC tasks have proven unexpectedly difficult for computational models, with only about half the tasks consistently solved on the private ARC-AGI-1 test set [Bober-Irizar and Banerjee, 2024, ARC Prize, 2024]. This difficulty emerges precisely because ARC demands *abstract generalization* over a minimal set of examples, thwarting superficial shortcuts. While ARC is not a perfect proxy for all human-level reasoning, it remains a valuable gauge of small-data adaptability, creative knowledge transfer, and flexible problem solving. The many caveats of ARC-AGI-1 were partially resolved in ARC-AGI-2 [Chollet, 2019, ARC Prize, 2025c]. As this second iteration exists only for a few months and not many custom approaches for this iteration exist

---

[1]For more detailed arguments, please see chapters I and II of Chollet [2019]

yet, we will primarily focus on ARC-AGI-1 in this paper. Nevertheless, the setting and therefore the conclusions are similar for ARC-AGI-2. In what follows, we leverage ARC as a testbed to motivate why **efficient generalization requires multi-component integration**. For more details on limits, alternatives and why we still choose ARC, see Appendix D.

## 2 Alternative Views

To the best of our knowledge, there is currently no real alternative view focused on *efficient* generalization. However, the next closest approaches might be **(a) Extended LLMs** (2.1) as a not-so-efficient but generalizing method, and **(b) Neuro-Symbolic** methods (2.2) as an efficient but not generality-focused approach.

### 2.1 Extended Large Language Models

Transformers and LLMs have undeniably exhibited broad emergent capabilities, including surprising generalization and few-shot reasoning, across multiple domains [Bubeck et al., 2023, Webb et al., 2023]. When *extended* with techniques like chain-of-thought prompting and test-time fine-tuning, they can perform competitively even on ARC [Greenblatt, 2024a, Berman, 2024, ARC Prize, 2025a]. With resource-heavy test-time optimization, models like GPT-4 and Sonnet 3.5 achieve up to $87.5\%$ on ARC-AGI-1, leading many to view LLMs as the foundation for future general-purpose AI[ARC Prize, 2025a].

**Strengths of LLMs.** Modern LLMs exhibit several key strengths. **Pre-training on Massive Corpora** allows for extensive self-supervised learning on diverse text sources. This way, LLMs acquire a wealth of representations, effectively consolidating and covering wide-ranging knowledge [Bubeck et al., 2023]. **Flexible Transfer of Knowledge** can be applied to handle various downstream tasks (including non-linguistic tasks expressed in language) with minimal fine-tuning, thanks to in-context learning, powerful embedding spaces, and diverse test-time strategies [Dong et al., 2023, Berman, 2024]. **Emergent Reasoning Behaviors** can be elicited through prompting strategies such as chain-of-thought or retrieval augmented generation. Such reasoning-like procedures within LLMs often improve the performance on complex tasks [Webb et al., 2023].

**Challenges and Limitations.** Despite impressive benchmark results, LLMs still exhibit significant hurdles regarding *efficient* generalization:

1. **Opaque and Brittle Emergence:** The extent to which LLMs can perform genuine abstract reasoning (versus pattern matching) remains an open debate [Valmeekam et al., 2023, Kaddour et al., 2023, Dziri et al., 2023, Lewis and Mitchell, 2024, Wang et al., 2024a, Lotfi et al., 2024, Schuurmans et al., 2024, ARC Prize, 2025f]. Their "emergent" abilities can be unreliable, hard to interpret, and domain-specific [Bober-Irizar and Banerjee, 2024]. For example, on ARC-AGI-1, the best performing LLMs achieve up to $87.5\%$ but on ARC-AGI-2 only $4\%$, while humans consistently reach $100\%$ [ARC Prize, 2025d].

2. **Data-Hungry and Costly:** Training large-scale transformers demands massive, human-generated corpora – and some fear we are reaching the upper limit of high-quality data for further scaling this approach [Sutskever, 2024]. In addition, fine-tuning and extensive resource-intensive test-time synthesis methods are expensive (money and time) [Sachdeva et al., 2024, Greenblatt, 2024a, Berman, 2024]. For example, on ARC-AGI-1, the amount of solved tasks **scales logarithmically** with the inference compute at test time [ARC Prize, 2025a]. For o3 to reach $75.7\%$ on the semi-private ARC-AGI-1, \$20 and $13.8$ minutes per task were necessary. To obtain $87.5\%$, roughly \$3400 and $3.7$ hours[2] were reported [ARC Prize, 2025e]. In comparison, an average human STEM graduate reaches $98\%$ requiring \$10 [ARC Prize, 2025d]. For further discussion on scaling efficiency, see Appendix H.3.

3. **Developer vs. Model Intelligence:** Many LLM-based successes rely heavily on *engineered prompting* and human-coded heuristics. The latest ARC-AGI-1 results reveal that while LLM-based approaches can outperform other methods on the public benchmark, they do so through *massive prompt engineering* [Greenblatt, 2024a, Berman, 2024]. Thus, high-level performance may reflect *developer-centric*[3] skill more than an intrinsic model capacity for

---

[2]Corrected for 173 times more compute compared to *low* setting
[3]Definition see Appendix B

generalization [Chollet, 2019, Dong et al., 2023, Yu et al., 2023, Bober-Irizar and Banerjee, 2024].

4. **Lack of Transparency:** Unlike modular designs, LLMs encode reasoning steps in vast weight matrices, limiting interpretability. This black-box nature impedes deeper analysis of the reasoning process and complicates improvements targeted at genuine compositional intelligence [Garcez and Lamb, 2023]. For more details, see Appendix A.

**Conclusion for LLMs.** Even though LLMs are powerful in practice, they do not generalize efficiently. Mahowald et al. [2024] draw parallels to the human brain's specialized "language areas," cautioning that forcing a language-dominant model to cover abstract non-linguistic tasks may be fundamentally inefficient. We cannot know yet if the continuing trend of extending purely neural LLMs with fancy strategies like Chain of Thought (CoT), In-Context Learning (ICL), Retrieval Augmented Generation (RAG), test-time compute, etc., will make them ultimately more efficient at generalization. This inefficiency might very well be fundamental, but we nevertheless encourage the continuing efforts to embed LLMs into neuro-symbolic frameworks. For a more detailed discussion, see Appendix H.3 and C.

## 2.2 Neuro-Symbolic Strategies

The term *neuro-symbolic* (sometimes abbreviated *NeSy*) can encompass a wide variety of hybrid architectures and learning strategies. While the specific mechanisms vary, the core idea is to marry *symbolic structures* (e.g., logic programs, Domain-Specific Languages (DSLs), knowledge graphs) with *neural components* (e.g., deep networks or learned embeddings) [Hitzler et al., 2022, Garcez and Lamb, 2023, Keber et al., 2024]. These two paradigms clearly complement each other [Bober-Irizar and Banerjee, 2024], already hinting at the potential of neuro-symbolic methods to tackle a broader range of tasks than each paradigm alone [Bober-Irizar and Banerjee, 2024, Chollet et al., 2025].

As multiple works have already surveyed the general advantages and disadvantages of neuro-symbolic approaches in depth [Hamilton et al., 2022, Hitzler et al., 2022, Garcez and Lamb, 2023, Keber et al., 2024, Bhuyan et al., 2024], we will not reiterate these existing arguments. Instead, we focus here on the key aspects *relevant for generalization*. Nevertheless, for completeness, we provide a brief discussion of symbolic approaches in Appendix G.

**Relevance in LLMs** Methods like chain-of-thought prompting and structured reasoning graphs already incorporate neuro-symbolic principles [Hitzler et al., 2022]. These techniques wrap neural transformers in symbolic scaffolding [Yu et al., 2023], improving performance across tasks. Examples include tree-of-thought [Yao et al., 2023] and graph-based reasoning [Besta et al., 2024]. Xu et al. [2024] demonstrate how such logical orchestration around LLM calls enhances reliability on diverse tasks. Consequently, also on ARC-AGI-1, the top LLM-based approaches incorporate symbolic heuristics to stabilize generalization [Franzen et al., 2024, Barbadillo, 2024, Chollet et al., 2025].

**Advantages and Successes** In recent years, researchers have increasingly aimed to harness the advantages of combining neural and symbolic paradigms[Hitzler et al., 2022, Garcez and Lamb, 2023]. That this strategy is fruitful could be shown by some first successes like the neuro-symbolic theorem prover AlphaGeometry [Trinh et al., 2024]. Also, the most recent ARC-AGI-1 findings [Chollet et al., 2025] show that neuro-symbolic approaches are a promising route to generalization. A clear illustration is Bober-Irizar and Banerjee [2024], who build upon a DSL-based ARC solver by adding learnable "concept formation" components, significantly boosting efficiency and success rates. Hybrid models can learn abstract concepts more compactly, leveraging both (i) a neural module to handle noisy or high-dimensional inputs and (ii) a symbolic module to enforce logical coherence and compositional reasoning. This synergy is particularly relevant in low-data tasks like ARC, where purely neural systems often overfit, and purely symbolic systems lack robust inductive priors. Table 3 in Appendix F summarizes a few representative state-of-the-art neuro-symbolic approaches that have been shown to be effective for generalization in ARC-like tasks.

**Challenges and Limitations** While recent work has demonstrated promising gains on ARC [Moskvichev et al., 2023, Chollet et al., 2025, Bober-Irizar and Banerjee, 2024], open challenges remain – most notably:

1. **Exploding Search Spaces.** Combining symbolic search with neural heuristics can mitigate the worst-case combinatorial complexity explosion, but designing these heuristics remains nontrivial [Bober-Irizar and Banerjee, 2024].

2. **Balancing Data Efficiency and Model Complexity.** For example, ARC tasks demand strong reasoning from minimal examples, stressing the importance of balanced architectures that do not over-parameterize [Moskvichev et al., 2023].
3. **Dynamic Concept Formation.** Handling ever-evolving domains requires neuro-symbolic methods that can *learn new concepts dynamically* rather than rely solely on a hard-coded DSL [Bober-Irizar and Banerjee, 2024].
4. **Underspecified methodology.** To the best of our knowledge, there is very little standardized methodology in the field of neuro-symbolic AI research. We are still searching for a consensus on the most effective ways to combine neural and symbolic approaches [Feldstein et al., 2024].
5. **Limited Focus on Generalization.** The promise of neuro-symbolic integration is as significant as it is unspecific. Combining the two major AI research paradigms of the last 80 years will open many possibilities; efficient generalization might be one of them, but not the prime focus of the strategy [Garcez and Lamb, 2023, Bhuyan et al., 2024].

**Conclusion for Neuro-Symbolic Approaches** Though the obstacles mentioned above are significant, the ability of neuro-symbolic methods to unify inductive and deductive reasoning is an especially potent strength – analogous to "System 1" vs. "System 2" thinking in human cognition [Kahneman, 2011, Garcez and Lamb, 2023]. As computational and data constraints grow more relevant, this marriage of neural and symbolic approaches will likely become unavoidable for efficient models. Unfortunately, the field of neuro-symbolic AI research is still in its infancy, quite underspecified regarding concrete methodology and not particularly focused on generalization, rendering it not particularly helpful for advances on skill-acquisition efficiency [Garcez and Lamb, 2023, Feldstein et al., 2024, Bhuyan et al., 2024].

## 2.3 Closing Remark

In summary, LLM-focused approaches can demonstrate remarkable capabilities but often rely on extensive engineering, computational resources, and data, with limited inherent interpretability and skill-acquisition efficiency. Neuro-symbolic methods stand at the intersection of statistical learning and explicit symbolic reasoning, promising synergy effects far beyond what either paradigm can achieve alone. They might be the most promising previously known route to efficient, transparent generalization. However, they only *hint* at the power of multi-concept integration and are underspecific when it comes to efficient generalization.

# 3 Pillars of Efficient Generalization

Based on historical examples, recent approaches, and models designed for the ARC benchmark, we propose that the following **fundamental pillars** are indispensable for attaining *efficient* generalization. A key insight of our work is the interrelation between these pillars, which we highlight by referencing connections between pillars using section references. For an overview, see Figure 1.

## 3.1 Model Specificity

Currently, we are unable to produce *efficient* models for general intelligence. For example, extended LLMs (see Section 2.1) generalize over a broad range of tasks but possess low *skill-acquisition efficiency* when accounting for all relevant factors (see Section 1). When hitting the practical limits of transformer scalability, the research community is forced to increase skill-acquisition efficiency to further improve generalization. The latest developments regarding reasoning models underpin this observation [DeepSeek-AI et al., 2025, Ballon et al., 2025].

We argue that the process should be the other way around: first achieving **high** skill-acquisition efficiency over a **limited scope**, which later needs to be extended, instead of aiming for **low** skill-acquisition efficiency over a **broad scope**, and later increasing the skill-acquisition efficiency. Consequently, a pillar for efficient generalization is properly fitting the model to the given scope. There are many steps in between *narrow task-specific AI* and *open-ended AGI*. We advocate for climbing this ladder slowly but thoroughly, aiming for high skill-acquisition ability always and only increasing the domain size (and difficulty).

For instance, in ARC, there is a defined set of known "core knowledge" priors (e.g. shape manipulation, counting, etc.), from which tasks are constructed. The puzzles seem relatively simple, but the

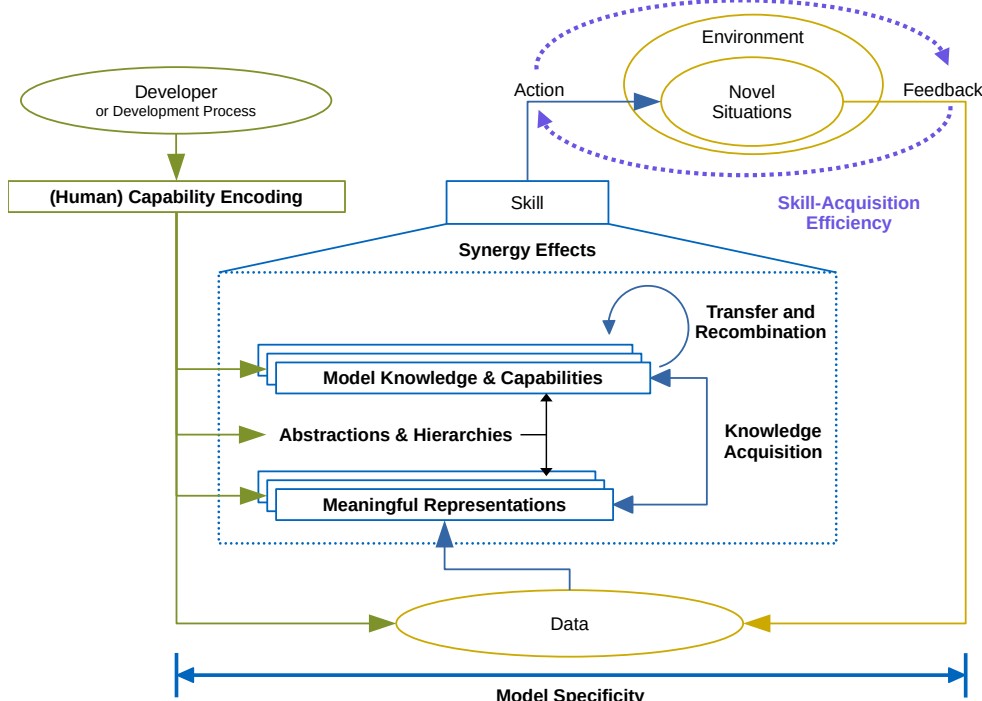

Figure 1: **Conceptual illustration of the six pillars for efficient generalization. Blue**: Components of a generalizing model. **Green**: Human development components. **Yellow**: Environment and Data. **Oval shapes**: Entities. **Bold**: Pillars (**Knowledge Acquisition** and **Transfer** separated for clarity). **Explanation**: The overall scope of the model, **data**, **development** is dictated by the **Model Specificity** (3.1). The **Developer** (or development process) is responsible for encoding **(Human) Capabilities** (3.2) into the model as well as preselecting data. The data is represented using **Meaningful Representations** (3.3). The model uses **Abstractions & Hierarchies** (3.4) to work at appropriate levels of granularity. On this basis, the model **extracts knowledge and competences** (3.5) which are reiterated over (**transfer** and **recombination**). Finally, all components of the model and the development process create **Multi-Component Synergy Effects** (3.6) that contribute to the model's **skill** level. The skill determines the **action** the model takes in a **novel situation** (which is determined by the **environment**). The situation will return some **feedback** signal, which is interpreted as new data. Based on this environment reaction, the model updates its inner states, improves its skill and (hopefully) generalizes to *the underlying task*. The (interactive) cycle between model action, environment reaction, model update and *skill improvement* is where we observe **skill-acquisition efficiency** in the end.

enormous flexibility of conceptual instantiations and recombination makes ARC hard to generalize over [Chollet, 2019, Ellis et al., 2020]. Efficiently solving this sub-challenge of intelligence allows for concentrating development efforts on generalization while preventing runaway complexity [Chollet et al., 2025].

By narrowing the domain, models can incorporate strong assumptions or exhaustive knowledge about that domain, yielding deeper generalization within the scope (at the expense of versatility). Here, lies a strong connection with the *no free lunch theorem* we discuss more extensively in the Appendix H.4.

**Takeaway**: A *targeted model scope*, with sufficient coverage of relevant key primitives yet focused capabilities, yields a broad solution space while still being feasible and tractable.

## 3.2 (Human) Capability Encoding

(Human) Capability Encoding focuses on how human competence can be injected into a system (e.g., via curated datasets, architectural biases, or manual hyperparameter tuning). It addresses what competences are pre-loaded into a model before training. Symbolic frameworks are predetermined

to incorporate human knowledge, but usually in a very rigid way, killing adaptability. For more details on generalization in symbolic systems, see appendix G. An illustrative example of how little generalization performance might be achieved despite extensive human knowledge encoding pose the top solutions to the initial ARC challenge in 2020 [icecuber, 2020, de Miquel, 2020, Larchenko, 2020].

Consequently, it is crucial to carefully focus on *abstract processes* (3.4), instead of hard-coding low-level solutions, when encoding human capabilities into a model. The idea is that the expensive extraction of high-level concepts from low-level data (3.5, 3.4) can partially be avoided making the overall systems significantly more efficient. Especially when it comes to absolute truths, such as moral (alignment), we do not want to learn from scratch, nor do we want a statistical modeling. Ellis et al. [2020, p 18] emphasizes that "rich systems of built-in knowledge" radically accelerate learning – a stance aligning with the principle that *broad* competence arises from fundamental, composable operators. For ARC-AGI-1 Xu et al. [2023a] showcases how rigid function definitions can be generalized by defining *process-level* abstractions (e.g. "move(object, vector)"), resulting in reusability across countless tasks as well as a significant reduction in search space.

**Takeaway**: Injecting *abstract human expertise* (concept-level rather than solution-level) boosts data efficiency and encourages flexible reuse.

### 3.3 Meaningful Representations

Meaningful Representations are the *internal model of reality*; they define the complexity and structure of the system's "world model", whether continuous embeddings or discrete symbols. They can be engineered (e.g., ontological graphs) or emerge purely data-driven (e.g., embedding spaces). Representation spaces influence how the environment is perceived and processed, ultimately shaping what abstractions and inferences are possible. While subject to capability encoding and abstractions, they are a distinct aspect/pillar of a system.

Representational design profoundly shapes a system's ability to generalize. While neural embeddings capture latent structure, they can be overly broad for specialized tasks like ARC (3.1) [Garcez and Lamb, 2023, Skean et al., 2024]. On the other hand, graph- or object-centric representations can simplify the model's action space. For example, Xu et al. [2023a] demonstrate how a simple shift from pixel based representations to object based representations can reduce search complexity by factor 10 while simultaneously increasing model interpretability. In contrast to abstraction capabilities (3.4), which are more processing-focused, meaningful representation spaces reflect the model's perspectives on the world (i.e., world model) [Huh et al., 2024, Barbadillo, 2024].

**Takeaway**: Accurately aligning representations with the *natural granularity* of the domain maintains computational efficiency while setting a meaningful scope for the model.

### 3.4 Abstractions and Hierarchies

Abstractions and hierarchies define the ability to transform raw input into progressively more conceptual representations by discarding irrelevant details. For example, a convolutional neural network, hierarchically abstracts raw pixels into high-level semantic features. This allows to process initially different input as similar at a higher level, which is central for generalization. The system recognizes structurally similar scenarios and might transfer learned skills more readily (3.5). Latest in with AlexNet, the significance of this approach became apparent as it was able to more robustly generalize over their respective image classification tasks [Alom et al., 2018]. This flexibility is a distinct phenomenon from merely encoding a competence (3.2) or having a fitting internal representation (3.3). Layered abstractions are foundational to both human cognition and deep-network architectures [LeCun et al., 1989, Riesenhuber and Poggio, 1999, Grill-Spector and Malach, 2004, Krizhevsky et al., 2017]. In the ARC-AGI-1 context, moving from pixel-level to object- or pattern-level operations delivers major efficiency improvements [Xu et al., 2023a,b]. Each abstracted layer or module discards noisy details, accentuating shared structures across tasks while bolstering interpretability.

**Takeaway**: *Hierarchical design* combines low-level perception and high-level logic, enabling compositional reasoning and meaningful explanations/representations.

### 3.5 Knowledge Acquisition, Transfer, and Combination

No matter how thorough the initial capability encoding, novel situations inevitably appear. Thus, an intelligent system must *learn* fresh concepts during training, convert them into something useful, and

*recombine* them spontaneously at inference time [Chollet, 2019]. It is debatable whether *knowledge acquisition* (how to get it) and *knowledge transfer* (how to use it) shall be considered two distinct pillars. Nevertheless, classical machine learning, primarily focuses on this pillar and has achieved major success so far.

**Knowledge acquisition** from data is fundamental for a generalizing system. A prerequisite are usually comprehensive training algorithms and fitting model architectures with sufficient capacity. Illustrative examples of the usefulness of knowledge acquisition is Transfer learning; for instance, a neural network pre-trained on a large dataset (like ImageNet or Wikipedia text) can be fine-tuned on a smaller target task, leveraging learned features or language understanding generalizing with less data [Weiss et al., 2016]. Also, multi-task learning utilizing shared representations (3.3) was shown to speed up the learning process [Zhang and Yang, 2018]. Multimodal LLMs follow a similar paradigm and are demonstrably applicable to many tasks/domains [Wang et al., 2024b]. Likewise, the field of meta-learning heavily relies on extracting useful information from the given data to generalize to new situations [Vettoruzzo et al., 2024].

The difference between knowledge acquisition and transfer is often fluid. However, **Knowledge transfer** from one domain to another is more involved; it requires systems to refine existing knowledge into concepts which are also useful on unseen tasks. The dependence on abstractions (3.4) is apparent, as non-abstracted knowledge is usually too situation-specific to be transferable. Knowledge transfer and recombination might happen during training as well as at inference time. For example, DreamCoder's "sleep-wake" cycle continuously refines a library of existing abstractions (dynamic concept synthesis) to transfer abstracted knowledge onto novel situations [Ellis et al., 2020]. Bober-Irizar and Banerjee [2024] demonstrates how DreamCoder can be adapted for successfully handling diverse ARC puzzles based on different underlying concepts. These approaches are primarily focused on transfer during training, not inference.

A good example for knowledge recombination during inference (as re-training/fine-tuning is too expensive) are LLM-based approaches. On ARC-AGI-1, test-time fine-tuning (Test-Time Fine-Tuning (TTFT)) has proven an essential tool for high-performing LLM-based models [Akyürek et al., 2024, Chollet et al., 2025]. The highest performing ARC-AGI-1 solution so far relies heavily on TTFT [ARC Prize, 2025a].

**Takeaway**: Flexible generalization arises from *continual concept formation* plus *dynamic adaptation* at test time.

## 3.6 Multi-Component Synergy Effects

Although some concepts are more critical, other "side problems" like uncertainty handling, and capability encoding are equally significant for broad-scope generalization [Bhuyan et al., 2024]. Many advanced methods overlook *at least one dimension* (e.g. using trivial transformations or underpowered representations), losing potential flexibility [Franzen et al., 2024, Berman, 2024]. In contrast, a systematic approach addresses each sub-component fostering powerful synergy effects between them [Garcez and Lamb, 2023].

A classical example is AlphaZero that uses a combination of Monte Carlo Tree Search (MCTS), a neural network, and a DSL to achieve superhuman performance in chess, shogi, and Go [Silver et al., 2017]. It achieves this by synergizing pillars: it uses a deep neural network to represent game states (3.3) and learn policies/value functions via self-play (3.5), but it also integrates an MCTS planner (3.4) and a capability encoding (3.2) in form of hard-coded rules of the game. The result is a system that acquires superhuman skill over multiple board games without extensively prepared training data, human expert data or handcrafted evaluations [Silver et al., 2017, McGrath et al., 2022].

On ARC, Bober-Irizar and Banerjee [2024] utilize a DreamCoder-inspired neuro-symbolic approach that significantly outperforms naive DSL search by leveraging richer learned representations (3.3) and heuristics (3.5). It integrates a symbolic program synthesis over a ARC-specific (3.1) domain specific language (3.2). Over-time it recognizes via a neural recognition model (3.3), usefull terms it abstracts (3.4) and adds to its knowledge base ()3.5) [Bober-Irizar and Banerjee, 2024, Ellis et al., 2020].

**Takeaway**: By *holistically* optimizing each component in the system, one transcends individual contributions and achieves system-wide synergy, enabling more capable and efficient generalization.

### 3.7 Concluding Remarks on the Pillars

Although these pillars are inherently interrelated, each highlights a distinct mechanism supporting or constraining a system's competence to generalize over novel tasks. Collectively, the six pillars constitute the blueprint for efficient generalization. When each is addressed deliberately and woven together cohesively, the resulting system achieves far more than either paradigm alone. We posit, therefore, that *fully engaging these pillars is indispensable* for the next leap in data-efficient, developer-aware, potentially interpretable, and efficient general AI.

## 4 Conclusion

In this position paper, we have argued that *efficient generalization requires the deliberate integration of six fundamental pillars* (see section 3). Our analysis of historical and contemporary AI approaches indicates that the pillars are not merely optional enhancements but *crucial components* for achieving skill-acquisition efficiency. Most promising approaches to generalization already implicitly optimize for these pillars to varying degrees. By making this framework *explicit*, we provide researchers with a conceptual design principle for developing more efficient generalizing systems. On this basis, we propose three priority research directions to advance efficient generalization:

- **Holistic Generalization Benchmarking**: More comprehensive test suites to systematically measure how effective systems implement all six pillars and convert their capacities into synergistic skill-acquisition. ARC-AGI-style benchmarks go a first step, but cannot control (yet) for crucial confounding factors like data augmentation (undermining low-data constraints), training effort (e.g., compute, time, grokking, etc.) or developer competence leakage (inflating system-centric capabilities). Besides operating conditions (cost, time, compute), there is no analysis of *how* models arrive at a conclusion/output. Additionally, real-world-focused datasets will be necessary to demonstrate practical applicability and foster adoption beyond academic contexts.
- **Pillar-Aware Architecture Design**: Create modular architectures with explicit interfaces between components representing different pillars/concepts, allowing researchers to systematically study their interactions. If we want to measure the quality of our pillars, we need to concretely distinguish and collect metrics on them.
- **Harvesting Synergy Effects**: Develop methods to analyze and track the dynamic sharing of information across pillars. We might not be able to understand the complex interactions within the human brain, but for artificial systems we have a chance at understanding how their pillars interrelate and the synergy effects emerge, enabling us to leverage synergies more effectively during both training and inference.

We also found that the pillars must be developed in concert, as their *interplay and synergy effects* are what enable systems to generalize efficiently across diverse tasks with minimal data and developer engineering. While individual pillars have been researched in isolation, systems optimizing only for one or two dimensions inevitably fall short of robust generalization. This explains why purely neural or purely symbolic approaches – and even many current neuro-symbolic systems – struggle with efficient skill acquisition. At least for neuro-symbolic approaches, it has already been acknowledged that *proper* modular integration is crucial yet technically daunting [Garcez and Lamb, 2023, Chollet et al., 2025]. Similar to neuro-symbolic AI, where the individual components are not novel in isolation (neural networks, symbolic approaches), our pillars are not fundamentally new either. However, acknowledging the significance of their interconnection and the resulting effects offers a fresh lens on the **essence of generalization**. This also connects to the modular-yet-highly-integrated functional organization of the human brain, where specialized modules work in concert to create flexible cognition [Kahneman, 2011]. To our knowledge, no prior work has systematically mapped these six pillars or argued for their collective necessity in enabling true skill-acquisition efficiency. This six-pillar approach offers a systematic way to evaluate and enhance existing architectures while guiding the development of novel ones.

We urge the research community to shift focus from monolithic approaches or partial implementations toward comprehensive architectures that deliberately integrate the pillars. While developing such systems presents significant engineering challenges, we believe this path offers the most promising route to achieving truly efficient general intelligence – systems that, in the long run, can flexibly adapt to novel situations with minimal resources while potentially remaining transparent, interpretable, and reliable.

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

## A Behaviorism vs. Internalism

A longstanding philosophical debate pertains to whether only *external* behavior matters (behaviorism) or whether the *internal* mechanisms of thought carry essential explanatory value (internalism). In contemporary machine learning, this tension appears as "functionality vs. interpretability" or "black-box vs. transparent systems." High-performing but opaque models – like many Large Language Models – demonstrate that achieving sophisticated outputs does not necessarily illuminate the process by which the model reasons [Hernández-Orallo, 2020, Schlangen, 2021].

As these systems are deployed in sensitive or high-stakes environments, interpretability and control become paramount [Hassija et al., 2024]. Post-hoc explanations often provide only a partial window into massive parameter spaces, leaving significant uncertainties about *why* a particular decision was reached [Kenny et al., 2021, Slack et al., 2021, Leemann et al., 2023, Rong et al., 2023]. By contrast, **inherent model transparency** – via symbolic modules, meaningful structured interfaces, or modular architectures – can yield more reliable comprehension of internal processes, facilitate debugging, and bolster trustworthiness. Consequently, we argue that *internalist* considerations should shape the development of any model that aspires to broader, more systematic intelligence.

## B Developer-Aware Generalization

Even when a model attains notable performance on a suite of tasks, it is crucial to distinguish between *intrinsic generalization* and *engineered solutions*.

Many recent successes hinge on massive data curation, architectural tuning, or manual injection of priors – leading to impressive system-centric results, but not necessarily reflecting a model's capacity to *autonomously* learn how to solve unseen tasks. A "**developer-aware**"[4] perspective on skill acquisition controls for these extra-human interventions [Chollet, 2019]. In contrast, a *developer-centric* measure includes both the developer's competence as well as the model's competence.

Without this distinction, higher benchmark scores might be misinterpreted as an increase in the system's general intelligence, while instead, only the development process was optimized (e.g., more optimal hyperparameters do not make the model design more fitting, but the performance improves). This even applies to generality-focused benchmarks such as the Abstraction and Reasoning Corpus.

## C LLM's worth for understanding intelligence

The LLMs mentioned in Section 2.1 are even less suitable as an *academic research framework* for understanding the *mechanisms* behind generalization, which are as unresolved as they are crucial. Large data combined with sufficient computing resources can brute force solutions, but they do not illuminate the core processes underlying abstract reasoning. For those interested in deeper interpretability, explainability, or developer-aware skill acquisition, *neuro-symbolic integration* might be indispensable.

## D ARC-AGI limitations and alternative benchmarks

Relying heavily on the ARC benchmark(s) often appears limiting; therefore, we will shortly discuss prominent limitations, how other popular benchmarks compare, and why we still conclude that ARC is the best choice.

### D.1 Limitation: Visual Data

A known caveat of ARC is that it "only" tests for **geometric problem solving via image data**. This potentially puts heavily text-oriented models at a disadvantage. Transformers are indeed very sensitive to the input/output prompting of ARC-style tasks [Greenblatt, 2024b]. Fortunately, the trend of foundation models goes towards more modality agnostic models (i.e., multi-modal models) [Li et al., 2024, Wang et al., 2024b]. Especially given the *the platonic representation hypothesis* by Huh

---

[4]As defined by Chollet [2019] p. 10: "Developer-aware generalization: this is the ability of a system [...], to handle situations that neither the system nor the developer of the system has encountered before." Further, "Note that 'developer-aware generalization' accounts for any prior knowledge that the developer of the system has injected into it."

846 et al. [2024], the concrete modality of data will lose its relevance for highly capable/universal models,
847 making the visual nature of ARC tasks irrelevant in the future.

## D.2 Limitation: Too artificial and not real-world enough

849 ARC is often perceived as overly structured or "toy-like." We argue this is not a limitation but
850 a hallmark of isolating the **gist of a problem**. ARC specifically tests for generalization while
851 minimizing all other factors. This does not mean that crucial properties are missing; for example,
852 there exist multiple ARC tasks specifically focused on ambiguity and noise removal.

853 In our opinion, "real-world" data does not have any additional properties relevant for generalization.
854 Similarly, Adam et al. [2019] came to the conclusion that the complexity of real-world tasks is highly
855 overestimated and the prevalent meta-structure in real data is heavily underestimated.

856 Furthermore, unspecified *noise may obscure* whether a model genuinely generalizes or simply
857 leverages massive data, shortcuts, or pattern memorization. ARC helps evaluate generalization in a
858 controlled setting, offering valuable insights before tackling noisy real-world environments. Given
859 the early research stage of skill-acquisition-focused approaches, an overly unstructured setting is
860 *counterproductive* for progress.

861 Furthermore, we believe that researchers unfamiliar with ARC tend to significantly **overestimate**
862 **the structure** in the actual ARC challenge. Between tasks, there is no structural similarity, resulting
863 in a lack of invariants between tasks (except the underlying geometric rules). If there were easily
864 exploitable structural patterns across ARC tasks, traditional neural networks and feature extractors
865 would work significantly better on ARC, but they do not. Consequently, on ARC-AGI-2, even LRMs
866 have major issues improving beyond a single-digit performance count [ARC Prize, 2025c].

867 While practical deployment requires **large-scale integration**, lessons from ARC can inform real-
868 world contexts where robust, data-efficient learning is critical. Consequently, the insights gained on
869 ARC are also valuable for the real world. Implementing them in real-world systems is primarily an
870 engineering challenge (and not the scope of this position paper).

## D.3 Alternatives

872 ARC is not the only dataset targeting higher-level reasoning, but it is the only one that explicitly
873 stresses *skill-acquisition efficiency*. For completeness' sake, we want to give some details on why
874 other, popular "reasoning" benchmarks are less suitable regarding efficient skill acquisition. Table 1
875 provides an overview of the most prominent alternatives (selection is not exhaustive and potentially
876 subjective).

| Benchmark | Core Skill Tested | Efficiency Gap |
|-----------|-------------------|----------------|
| **ARC** | Geometric problem solving, abstract reasoning | Explicitly designed to measure skill-acquisition efficiency. On ARC-AGI-2, Humans: 98%, LLMs : <4% [ARC Prize, 2025d] |
| **BIG-Bench** | Mixed reasoning, code, riddles | Many-shot CoT sampling lets models brute-force via scale [Srivastava et al., 2023] |
| **MMLU** | Broad domain knowledge | Scores dominated by pre-training, not in-task learning [Hendrycks et al., 2021] |
| **SciEval** | Dynamic STEM QA | Unlimited query passes hide inference compute cost [Sun et al., 2024] |
| **GPQA** | Expert closed-book retrieval | Measures "knows," not "learns"; retrieval > abstraction [Rein et al., 2023] |

Table 1: Popular reasoning benchmarks versus *skill-acquisition efficiency*. Each still rewards brute-force scale or developer engineering, obscuring the cost–competence ratio that ARC measures.

877 We shortly describe each benchmark and where we think they are lacking:

- 878 • **BIG-Bench** (Beyond the Imitation Game Benchmark), is a collaborative benchmark with
  879 200+ tasks such as code debugging, riddles, and obscure language translation. It is supposed
  880 to test robustness to ambiguity and compositional reasoning through diverse challenges
  881 requiring logic, creativity, and cross-domain knowledge transfer [Srivastava et al., 2023].

It includes many complex reasoning tasks and even "challenge" tasks beyond current AI, some of which require reasoning steps or creative generalization, testing *flexibility* to an extent. However, models are evaluated in a zero- or few-shot prompt setting, and many tasks still correlate with knowledge or patterns seen in training. Indeed, tasks with a large knowledge component show gradual improvement as model size increases (suggesting *pattern-learning*), whereas truly novel multi-step reasoning tasks remain unsolved until models hit a scale "breakthrough" point. In short, BIG-Bench does probe generalization, but it does not uniformly enforce *learning new skills* – some tasks can be partially solved by pattern matching or sheer pre-trained knowledge [Srivastava et al., 2023].

- **MMLU** (Massive Multitask Language Understanding), covers 57 subjects across STEM, humanities, and social sciences to assess broad knowledge application. It is supposed to measure how models generalize across disciplines rather than excelling at narrow tasks [Hendrycks et al., 2021]. MMLU is knowledge-centric and less focused on new skill learning. It primarily measures factual knowledge and some reasoning acquired from large-scale training. There is little focus on adapting to *novel* tasks as questions resemble those in textbooks or exams. Strong model performance often comes from more training data or model parameters, not *on-the-fly abstraction*. Additionally, MMLU seems to suffer from dataset quality and transparency issues [Gema et al., 2024].

- **SciEval** (Scientific Evaluation Benchmark), uses dynamically generated questions in physics, chemistry, and biology to prevent memorization. Models shall demonstrate genuine scientific reasoning, with GPT-4's accuracy dropping from 65% to 26% when tested on novel dynamic data [Sun et al., 2024]. It is valuable for assessing in-depth reasoning in science, requiring models to apply scientific knowledge and logic rather than just memorize answers. The inclusion of dynamically generated problems means models face some novel content, revealing whether they can reason beyond rote memory (GPT-4 still has "substantial room for improvement" on these dynamic questions [Sun et al., 2024]). However, SciEval is limited to the scientific domain – it measures generalization *within* science (e.g., applying known principles in new ways) more than general skill acquisition across arbitrary tasks. Models are not learning entirely new kinds of tasks; they are answering science questions (albeit challenging ones) using prior scientific knowledge. This is a test of *expertise* and *reasoning depth*, but not *learning to learn* broadly outside the science context [Sun et al., 2024].

- **GPQA** (Graduate-Level Google-Proof QA), features 448 expert-level multiple-choice questions in biology, physics, and chemistry where internet access provides minimal human performance gains (34% vs 25% baseline). This benchmark tests for deep conceptual understanding rather than information retrieval capabilities [Rein et al., 2023]. It specifically targets questions that demand (in humans) complex reasoning or deep understanding, beyond simple fact recall. It highlights the gap between human experts and AI on truly *hard* questions. In terms of skill acquisition, though, GPQA remains a static question benchmark. The model is not asked to *learn* a new skill; it is challenged to apply its existing high-level knowledge in novel, intricate ways. Failing GPQA often means the AI lacks the necessary combined knowledge or reasoning chain, rather than failing to learn from small data (since no new training is given). Thus, GPQA is excellent for stress-testing an AI's reasoning within known domains, but it does not evaluate the process of *rapidly learning* an entirely new type of task or concept [Rein et al., 2023].

Most benchmarks **assume** that some aspect of human intelligence is required to perform well on the given dataset, simply because humans would require intelligence to solve it. A theoretical foundation for such assumptions is usually lacking. Historically, *chess* was also once considered a real-world intelligence benchmark as humans require a diverse repertoire of skills to solve it (tactics, reasoning, planning, means-end analysis, theory of mind, deception, etc.) [Newell, 1973]. However, it later became obvious that a naive tree-search method - lacking all signs of intelligence - is perfectly capable of solving chess. We are therefore cautious not to overestimate the value of "established" benchmarks of intelligent agents that do not have an explicit grounding regarding skill-acquisition efficiency.

To cite the ARC foundation themselves (ARC Prize [2025c]):

> *All other AI benchmarks focus on superhuman capabilities or specialized knowledge by testing 'PhD++' skills. ARC-AGI is the only benchmark that takes the opposite design choice – by focusing on tasks that are relatively easy for humans,*

> *yet hard, or impossible, for AI, we shine a spotlight on capability gaps that do not*
> *spontaneously emerge from "scaling up".*

As Chollet [2019] substantiated ARC with an extensive theory of how to measure generalization, we find the ARC challenge to be much more valuable for progress on AGI than any currently existing *"real-world"* benchmark. Of course, we are aware that this claim is up for discussion and remains to be proven right or wrong with time.

### D.4   Conclusion

ARC it is one of the few benchmarks explicitly designed to minimize statistical shortcuts and **emphasize skill-acquisition efficiency**. By removing extraneous noise and avoiding easily exploitable patterns, ARC isolates how effectively a model can acquire and transfer skills from limited experience. Pinpointing and measuring skill acquisition efficiency is central to generalization (see 1) and a key research question in itself. To the best of our knowledge, ARC is the most relevant benchmark regarding this notion of generalization, which is why it's our priority.

While ARC is not perfect, it provides an environment to observe the interplay of multiple factors (architectural choices, knowledge priors, hierarchical abstractions, etc.), providing an ideal testbed for novel generalization approaches, fostering progress.

## E   Dimensions for designing models

Certain system properties (e.g., transparency or interpretability) are not strictly required for a model to generalize well on some tasks. However, as discussed in Section A, one should focus beyond mere performance. To offer design guidance for more robust, reliable, and maintainable systems we provide guardrails for the developmental process in Table 2. These are suggested design principles rather than mandatory criteria.

Table 2: Key Dimensions for Designing Models with Broad Generalization

| Dimension | Importance for General Intelligence | Representative Works |
|---|---|---|
| **Skill-Acquisition Efficiency** | Emphasizes how well a system adapts to new tasks without extensive retraining; penalizes overreliance on developer engineering or huge datasets. | Chollet [2019], Bober-Irizar and Banerjee [2024] |
| **Transparency & Interpretability** | Strengthens trust and debugging; post-hoc explanations are often insufficient for large black-box models. Inherent transparency is crucial for real-world reliability. | Hernández-Orallo [2020], Hassija et al. [2024] |
| **Symbolic Reasoning** | Allows compositional, logically coherent transformations. Fosters human-level abstraction and provides robust handling of discrete structures. | d'Avila Garcez and Lamb [2023], Keber et al. [2024] |
| **Neural Representations** | Harnesses powerful pattern-extraction capabilities from raw data (images, text), enabling feature discovery and capturing nuanced correlations. | Bubeck et al. [2023] |
| **Small-Data Adaptation** | Avoids brute-forcing solutions by demanding strong generalization from very few examples (as in ARC tasks), exposing true abstraction capabilities. | Moskvichev et al. [2023], Chollet et al. [2025] |

## F   Representative Neuro-Symbolic Approaches

Table 3 summarizes representative state-of-the-art neuro-symbolic approaches that have been shown to be effective for generalization in ARC-like tasks.

## G   Purely Symbolic Approaches: Domain-Specific Languages and Program Synthesis

Although overshadowed by neural methods in recent years, purely symbolic or logic-based AI once dominated AI research and retains a devoted following Kastner and Hong [1984]. To provide an overview of their limitations for current generalization challenges we analyze them in context of

Table 3: Representative Neuro-Symbolic Approaches for Generalization in ARC-like Tasks

| Approach | Neural Component | Symbolic Component | Key Mechanism & Insights |
|---|---|---|---|
| **Bober-Irizar & Banerjee (2024) [Bober-Irizar and Banerjee, 2024]** | Learned concept-formation module (e.g., CNN-like embeddings to identify object features) | DSL-based program search for transformations | Uses neural heuristics to guide symbolic search, significantly reducing the DSL's combinatorial explosion. Demonstrates notable gains on ARC tasks versus purely symbolic baselines. |
| **DreamCoder [Ellis et al., 2020]** | Neural "wake-sleep" cycle that learns common subroutines or concepts | Inductive program synthesis in a high-level language (with control-flow, recursion) | Iteratively refines a library of reusable functions – symbolic *abstractions* – guided by neural scoring. *DreamCoder* is not specifically designed for ARC but illustrates how learned domain knowledge can be symbolically encoded. |
| **Neuro-Symbolic DSL Enhancements (various) [Hamilton et al., 2022, Hitzler et al., 2022, Garcez and Lamb, 2023, Bhuyan et al., 2024]** | Neural embeddings for object detection, classification, or spatial feature extraction | Logic-based DSL or ontology enforcing compositional rules | General family of hybrid methods: neural modules handle perceptual tasks or fuzzy matches, while symbolic DSL enforces interpretability and constraint satisfaction. Shown to improve data-efficiency and interpretability on small "grid-world" or ARC-like puzzles. |

ARC-AGI-1. Within the ARC domain, the most visible symbolic attempts revolve around exhaustive search in a *Domain-Specific Language* (DSL) or program-synthesis methods such as DreamCoder Ellis et al. [2020].

**DSL-Based Methods.** Early top-ranked solutions in the original ARC challenge relied on large, hand-crafted DSLs icecuber [2020], de Miquel [2020], Larchenko [2020]. By systematically searching over a predefined set of transformations and heuristics, these approaches found valid transformations for specific puzzles. However, these DSL-based methods achieved only modest coverage due to the combinatorial explosion of possible transformations and the diversity of ARC tasks. They also demanded extensive human engineering to hard-code each concept, undermining *developer-aware* generalization measures Bober-Irizar and Banerjee [2024].

**Program Synthesis Approaches.** Program-synthesis frameworks like DreamCoder Ellis et al. [2020] extend the DSL idea with higher-level constructs (e.g., control-flow operators, recursion). While this unlocks greater expressiveness, it can also inflate the search space. Adapting a fully general programming language for ARC tasks becomes cumbersome because ARC-AGI-1 is already quite challenging without further increasing the solution space Bober-Irizar and Banerjee [2024].

**Symbolic Drawbacks.** While symbolic approaches can offer strong interpretability (one can often track each logical step explicitly), they typically struggle to infer abstract "core concepts" from limited data without some learned inductive biases. Their purely top-down logic has trouble coping with the noisy, high-dimensional input distributions where data-driven feature extraction is crucial. Additionally, naive symbolic search tends to be fragile in the face of tasks requiring approximate or probabilistic reasoning. A ubiquitous problem of symbolic approaches - even outside of ARC - is the above-mentioned complexity explosion, making scalability to real-world settings often infeasible [Garcez and Lamb, 2023].

**Conclusion** Historically, purely symbolic solutions have rarely scaled well across diverse tasks and have difficulty encoding robust priors for low-data settings Kastner and Hong [1984], Ellis et al. [2020]. Conversely, the golden era of symbolic AI faded in the late 1980s, giving way to sub-symbolic (neural) approaches. Still, the ARC challenge confirms that exhaustive or highly engineered symbolic DSLs rapidly reach diminishing returns. Hence, purely symbolic approaches, while valuable for interpretability and logic, alone are still insufficient for broad or efficient generalization.

The limitations that once suffocated symbolic AI – such as brittle rule systems or exponential search complexity – can be mitigated by modern neural advances and computing power Mira [2008]. However, those hybrid, neuro-symbolic approaches go beyond what we consider purely symbolic.

# H Frequently Asked Questions

## H.1 General Paper Structure and Approach

**Your paper reads like a survey, are you sure it's a position paper?** Yes. We intentionally include numerous references to prior literature as necessary grounding for our claims. Much research about AGI is (highly) speculative. Therefore, we carefully selected many different sources to substantiate our claims. Nevertheless, we are not only stating previous developments, but also distill a structured perspective on how to possibly achieve efficient generalization, which we would like the research community to consider more intensively.

**This is just a general description of desiderata. Why is it so vague?** We take a meta-level perspective, aiming to shift focus toward systematically addressing six core components (see Section 3) of generalization. We aim to articulate a conceptual framework of guardrails and design guidelines relevant for achieving robust skill-acquisition and generalization. Concrete work implementing multiple pillars to varying degrees already exists. Therefore, our motivation is to guide researchers toward systematically and consciously using these components to enable synergy effects resulting in effective generalization. Our paper shall broaden the conversation from "LLMs vs. Neuro-Symbolic" to "How do we achieve efficient generalization, and what is essential to that end?". We contribute a mindset shift for the research community (research direction), not concrete experimental results/insights.

Consequently, it is up to the ML community to further conceive, test, and refine systems specifically focused on these pillars. We provide a design paradigm that must be instantiated and therefore do not provide extensive algorithmic details, which would be more appropriate for a concrete research-track paper.

**Why focus on ARC when it's just a visual reasoning benchmark?** We address this more extensively in Appendix D. In short, ARC is not merely a visual reasoning benchmark but a carefully designed test for generalization capability that minimizes statistical shortcuts and emphasizes skill-acquisition efficiency. The visual modality is incidental, not central – what matters is the benchmark's ability to isolate generalization from other confounding factors. With the trend toward more modality-agnostic models and the Platonic Representation Hypothesis [Huh et al., 2024], the specific modality becomes increasingly irrelevant for highly capable models anyway.

## H.2 Six Pillars Framework

**How do your six pillars differ from neuro-symbolic frameworks?** While neuro-symbolic approaches typically focus on combining neural and symbolic components, our six-pillar framework takes a more comprehensive view. Existing frameworks tend to emphasize the neural-symbolic integration itself, whereas our pillars address multiple orthogonal aspects of generalization: from model specificity and (human) capability encoding to meaningful representation spaces and abstract hierarchies. Most of the pillars can be implemented either in a purely neuronal or purely symbolic way (even if that does not make much sense in practice).

Our framework also explicitly highlights the synergy effects between these components in terms of skill-acquisition efficiency rather than just performance or capability, prioritizing generalization from minimal data and experience.

**What concrete evidence supports the necessity of all six pillars?** The need for all six pillars is supported by analyzing the limitations of current approaches that excel in some areas but fail in others. For instance, purely neural approaches lack compositional reasoning (addressed by Multi-Component Synergy), while purely symbolic methods struggle with meaningful representation learning and adaptation (addressed by Knowledge Acquisition & Transfer). The success of approaches incorporating multiple pillars, such as the neuro-symbolic method by Bober-Irizar and Banerjee [2024], provides empirical evidence for their collective importance. However, to the best of our knowledge, no single system has yet fully optimized all six pillars simultaneously, which represents a key research frontier. Each pillar addresses distinct failure modes observed in existing AI systems when confronted with generalization challenges.

**How do you measure "efficiency" in skill acquisition?** We adopt the framework proposed by Chollet [2019], where efficiency is measured as the ratio between the competence gained and the resources required. Specifically, we consider: (1) data efficiency – how much performance is achieved from minimal examples; (2) computational efficiency – the processing resources needed during both

training and inference; (3) developer effort – controlling for human engineering in the system design; and (4) transfer capacity – how well skills learned in one context apply to novel scenarios. That the ARC benchmark (organizers) cannot control all the relevant factors (i.e., data augmentation, developer effort, training compute) is an open issue/limitation, which - however - applies to most benchmarks.

In an optimal world, an efficient system would solve novel ARC tasks with minimal resources while a less efficient system (like current LLMs) might achieve similar performance but at vastly greater computational or data costs. Chollet [2019] also proposed some formalisms how to calculate the skill-aquisition efficiency and mathematically correct for confounding factors (see Section II.2.2 of [Chollet, 2019]).

## H.3 LLMs and Scaling

**For AGI, why don't we just use LLMs, and make them more powerful? You already suggested extending LLMs with symbolic scaffolding to improve their lack of reasoning.** You can do that, and you will probably also get something that looks like AGI at some point. But that has not much to do with skill-acquisition efficiency. Making these approaches more efficient afterwards might be very hard, so we suggest we start with efficiently generalizing systems first and steadily expand the scope.

**We will make LLMs more efficient at some point. Computers were also building-sized and now are small.** Maybe you can increase the efficiency of a LLM-based system by the same order of magnitude as hardware improves (see Moore's law). But maybe we cannot, as there might be more fundamental issues. We will see with time. In the meantime, our position paper proposes a parallel research direction that is inherently focused on generality by design from the start.

As a reference, until ARC-AGI-1 could be solved with near-human performance, it required Large Reasoning Models. These models are currently at a low two-digit percentage on ARC-AGI-2 [ARC Prize, 2025c,b]. It will take some time (and effort) to crack ARC-AGI-2, not even mentioning ARC-AGI-3, which is also on its way [ARC Prize, 2025c]. This being said; ARC only tackles relatively straightforward geometric problems; there is no significant increase in domain/concept coverage between these increments of the ARC challenge, only the required adaptability and efficiency increases.

## H.4 Generalization

**There are indications that large, unspecific models are actually very good at generalization. Why then focus on the "model specificity" pillar?** Researchers like Goldblum et al. [2024] have indeed observed that LLMs (or more generally, "overparameterized" neural networks) do not tend to overfit as much as originally thought but rather generalize over topics. However, they also claim that this effect is related to the low Kolmogorov complexity of real-world data. They therefore argue (and we agree) that currently, the *no free lunch* theorems have little relevance for SoTA LLMs. Acknowledging that LLMs work and generalize does not make them efficient at doing so. So this does not help us much for skill-acquisition efficiency. Datasets like ARC, which are precisely focused on skill acquisition efficiency, do not possess the same low complexity/flexibility as real-world data. As a consequence, custom under-specific ARC solvers have a much harder time performing on ARC tasks. We therefore think that *efficient generalization* indeed is scope-dependent, making the *no free lunch* theorems relevant again.

On that note: human intelligence is not universal either. We are relatively optimized/specialized for the specific physical world we live in. The concept of *universal general intelligence* exists, and humanity is pretty far away from it [Everitt and Hutter, 2018].

