# OpenReview forum: "The Pillars of Skill-Acquisition and Generalization; Why efficient General Intelligence requires Multi-Component Integration"
_NeurIPS.cc/2025/Position_Paper_Track — Submitted to NeurIPS 2025 Position Paper Track_

### Official Review · Reviewer_iA6X · 2025-08-04

**Significance:** 2
**Presentation:** 1
**Rating:** 2
**Confidence:** 4

**Summary:**

This paper advocates for neurosymbolic AI, arguing that AI systems that can efficiently generalize need to use neurosymbolic AI to be successful. They list six pillars of for efficient generalization, based off the ARC-AGI benchmark.

**Strengths:**

The paper discusses an important topic — efficient generalization of AI methods — and issues related to an influential benchmark, ARC-AGI.

**Weaknesses:**

The argument is not clear. Terms are not clearly defined. Overall, the writing quality is the main thing dragging down the paper. The argument could be improved by making more precise terminology and arguments. For example, the paper proposes "multi-component integration" which could really mean anything in the context of AI architectures. Similarly, another pillar is "knowledge acquisition, transfer, and combination" which is vague.

The paper discusses lots of interesting topics, and the discussion about each topic is interesting, but the overall argument is muddled. It mentions many different interesting research topics related to neurosymbolic AI, but seems to be advocating for the field of research as a whole, while also trying to argue that the whole thing is necessary.

**Questions:**

If the position in your paper were to be agreed with and adopted by AI researchers, what changes would you see in the community? Are there any existing works which would be approached differently as a result of your position?

**Alternative Position:**

Yes, and alternative positions are well-considered and named but not addressed

**Author Identification:**

No.

**Context:**

3

**Discussion:**

1

**Ethics:**

["NO or VERY MINOR ethics concerns only"]

**Position:**

Yes, the paper argues for or against a position related to machine learning.

**Support:**

1

**Thoroughness:**

2

---

### Official Review · Reviewer_G3oW · 2025-08-06

**Significance:** 2
**Presentation:** 3
**Rating:** 3
**Confidence:** 3

**Summary:**

This paper proposes a few components that underpins the generalizability of AGI. Following Chollet [2019] it defines AGI as generalizability (skill-acquisition efficiency) , and uses  ARC generalization benchmark as the corresponding metric. It first reviews the limitations of existing neuro (LLMs) and symbolic approaches, and then lists main limitations of the combined neuro-symbolic approaches.
1) exponential search space
2) bad data efficiency
3) require hard-coded DSL for new concepts
4) lack of standard implementation

Then it proposes to extend neuro-symbolic approaches with 6 components.
1) A limited model scope (e.g., counting, shape manipulation) allows high skill-acquisition efficiency
2)  encode human knowledge at an abstract level (e.g. move(object, vector)) to avoid overfitting to details.
3) Embedding spaces can be lack of specificity, while graph/object representations are both interpretable and reduce search spaces compared to pixels.
4) Abstractions and Hierarchies are useful as demonstrated by conv nets and AlexNet.
5) Knowledge Acquisition, Transfer, and Combination (pretraining, fine tuning, test time fine tuning)

**Strengths:**

This paper summarizes recent findings related to the ARC benchmark.

**Weaknesses:**

My main concern with this paper is its lack of novelty. The listed limitations of neuro-symbolic approaches are mainly known limitations of symbolic approaches. The listed components are useful characteristics but none is particularly novel given research in the past decades. The components feel more like a summary of recent results in deep learning and the ARC benchmark, and they lack a novel perspective or hypothesis. The completeness of these components is also not discussed, because there is little evidence that they are the most important ones. For example, are long term memory and spatial representations also important for AGI?

**Questions:**

See weakness

**Alternative Position:**

Yes, and alternative positions are trivial straw-man arguments

**Author Identification:**

No.

**Context:**

3

**Discussion:**

2

**Ethics:**

["NO or VERY MINOR ethics concerns only"]

**Position:**

No, the paper presents new research without clearly advocating a position.

**Support:**

2

**Thoroughness:**

4

---

### Official Review · Reviewer_UWVi · 2025-08-11

**Significance:** 2
**Presentation:** 2
**Rating:** 4
**Confidence:** 3

**Summary:**

This paper adopts the definition of AI as skill acquisition efficiency, and advocates evaluation of a model's capacity to abstract knowledge from sparse data and adapt to novel tasks. This paper critiques current predominant research in Large Reasoning Models and neuro-symbolic approaches, arguing that they are insufficient to build a truly generalizable system.

Instead, they posit that a multi-component implementation strategy, which consists of six pillars, is necessary for efficient general intelligence. The six pillars include Model Specificity, (Human) Capability Encoding, Meaningful Representations, Knowledge Acquisition & Transfer, Abstractions & Hierarchies, and Multi-Component Synergy.

**Strengths:**

- The paper addresses a timely and important topic in the ongoing debate between neuro-based and symbolic methods, and identifies several valuable components for designing systems that are efficient in skill acquisition.

- The work is well-motivated from the perspectives of generalization and skill acquisition, and it clearly articulates both the advances and the limitations of recent LRM- and symbolic-based approaches.

- The discussion of the six pillars is well-grounded, with each point supported by relevant evidence from the literature.

**Weaknesses:**

- While the six pillars advocated are valuable, it is not clear whether they are strictly necessary for building such a system. Incorporating all six may increase system complexity, potentially hindering implementation and evaluation. The paper also lacks actionable guidance on the implementation of these pillars in real applications.
- The paper mentions generalization multiple times. However, the type or degree of generalization intended is not crystal clear. For instance, "local forms of generalization" (line 21) is not clearly defined. It would be clearer with precise definition and demonstrative examples.
- The evaluation centers on the ARC dataset, yet it is not clear how the six pillars apply to other datasets or domains. Many datasets may not easily be aligned with the structured style of ARC.

**Questions:**

- Instead of integrating symbolic modules with neural models at the architectural level, some approaches train neural models on neural–symbolic datasets (e.g., [1, 2]) in domains such as math, coding, and other symbolic tasks, showing improved performance and generalization on downstream tasks. How do you view this data-driven form of synergy between neural and symbolic methods, and how might it compare to the framework you discuss?\
[1] Math for AI: On the Generalization of Learning Mathematical Problem Solving\
[2] How Does Code Pretraining Affect Language Model Task Performance?

Would requiring all six pillars introduce unnecessary complexity into system design? For instance, some pillars are motivated by interpretability considerations, which can be challenging to achieve robustly. How practical and actionable do you consider the proposed pillars in real-world implementations?

**Alternative Position:**

Yes, and alternative positions are well-considered and named but not addressed

**Author Identification:**

No.

**Context:**

3

**Discussion:**

3

**Ethics:**

["NO or VERY MINOR ethics concerns only"]

**Position:**

Yes, the paper argues for or against a position related to machine learning.

**Support:**

3

**Thoroughness:**

3

---

### Note · Authors · 2025-09-04

**1-10 Additional Comments:**

Terrible reviews.
We submitted a first draft of your paper to ICML before, and since then, we have objectively improved the paper by a lot, but the reviews here at NeurIPS were even worse. None of the reviewers seems to have properly read and understood our paper. They also did not seem to understand what a position track is about in the first place.
And yes, we are certain that this is not an intrinsic issue of the paper because we were getting much more positive feedback from colleagues outside of this peer review process already.

**1-11 Submit Again:**

Definitely no

**1-1 Submission Process:**

3

**1-2 Next Year:**

- More rebuttal space. 2000 characters are insufficient to tackle all concerns raised by reviewers in the required depth.

**1-3 Future Development:**

- Proper rebuttal phase. Position papers are notoriously subjective and need some argumentation if reviewers do not agree or understand a position. The discussion phase should be at least as extensive as with the main track.
-  Proper reviewer briefing on how to review position papers. In my group, we also had one person reviewing NeuIPS position papers, and they told me that it's much harder than main track papers because it's less standardized. From the reviews we got on our own submission, we noticed that many reviewers stick to old main-track habits for reviewing ("Where are the Experiments?", "What about practical applicability?", etc.). None of our reviewers seems to have an understanding of what a position paper actually is supposed to be (they did not even know/implement things from the Call for Position papers when reviewing...)

**1-4 Interest:**

["Panel discussions with other position paper authors", "Structured debates on controversial topics"]

**1-4 Other Interest:**

- I would be interested in reviewing for the position paper track myself, as there seems to be a lack of people with the necessary skillset and knowledge about my topic. Unfortunately, given that these people are currently reviewing my own work, I get blocked from actually publishing myself...

**1-5 Thoughtful:**

2

**1-6 Supportive:**

3

**1-7 Technical Aspects Versus Position:**

2

**1-8 Gate Keeping:**

4

**1-9 Camera Ready Changes:**

- Include definition and distinction for local and broad generalization (as pointed out by reviewer UWVi)

**3-1 Review Response1:**

UWVi

**3-2 Reaction To Review1:**

- Regarding the necessity of pillars:
As of now, we can not be certain what is strictly necessary. We acknowledge this weakness, which is a matter for future investigation, being the overall point of this **position paper**. Nevertheless, we believe presented convincing evidence from SoTA models substantiating our pillar choice and their future research relevance.

- Regarding clarity of generalization definitions (including local and broad):
We adopt the definitions from Chollet 2019, which are extensively described in his original work (even providing formulas). To avoid confusion, we will include respective references/definitions in our final paper version.

- Regarding the role of ARC in our paper:
ARC is not a dataset benchmarking geometry skills, but generalization. To our knowledge, it is currently the most SoTA benchmark for generalization. Results and insight gained on ARC are transferable to other generalization problems. Analogy: The first breakthrough of CNNs was on digit classification (MNIST dataset), but nobody considers CNNs to be "digit classifiers". The architectural insights gained are much more fundamental and applicable to other domains. Dataset choice is secondary for the field of generalization. For more, see Appendix D.

- Regarding the pillars' complexity (reviewer question):
Unfortunately, we think the six pillars are the minimal amount of complexity required to achieve efficient generalization. Efficient generalization is no easy feat and took millions of years of evolution to emerge in biology. Some amount of complexity can be expected.

- We would love to discuss the other very interesting reviewer question here to but are limited by space :(

- Regarding actionable implementation details and the paper's ranking:
We strongly urge the reviewer to re-read the first few paragraphs of the NeurIPS 2025 Call for Position Papers ( https://neurips.cc/Conferences/2025/CallForPositionPapers ) and reconsider the evaluation of our paper.

**3-3 Review Response2:**

G3oW

**3-4 Reaction To Review2:**

- Regarding the summary:
The reviewer writes "[the paper] proposes to extend neuro-symbolic approaches with 6 components," indicating that they neither completely read nor understood the paper. We propose an *alternative* view to NeSy approaches, see lines 213-216, 223-224, 226-226 and 1032-1041

- Regarding the novelty of our pillars:
Indeed, none of the presented facts are fundamentally new (otherwise, we would just be speculating like many high-level AGI publications are).  Nevertheless;
a) Having insights from an actual generalization benchmark is new (ARC challenge results are from December 2024)
b) Our novelty/contribution lies in fitting together the puzzle pieces into one holistic perspective on efficient generalization and painting a picture of an underrepresented research area dealing with synergy effects. To the best of our knowledge, no author has done that before, especially with the breadth and depth of supporting historical and SoTA references.

- Why exactly these 6 pillars:
We cannot be 100% certain that we have not missed something, and whether these pillars are Mutually Exclusive, Collectively Exhaustive (MECE). However, we have provided convincing evidence that they are fundamental aspects of generalizing approaches, and their synergy effects should therefore be researched more thoroughly. Check line 1042 ff. for more details.

- Regarding long-term memory and spatial representations:
Specific pillar sub-categories are omitted in this work, as there are many ways to instantiate them. Are more fine-grained taxonomy is scope for future work. "Long-term memory" is under pillar 3.5 (model knowledge is, see Figure 1). "Spatial representations" a subproblem of "3.3 Meaningful Representations".

- Regarding everything else:
We strongly urge the reviewer to re-read our paper and the first few paragraphs of the NeurIPS 2025 Call for Position Papers ( https://neurips.cc/Conferences/2025/CallForPositionPapers ) and reconsider the evaluation of our paper.

**3-5 Review Response3:**

iA6X

**3-6 Reaction To Review3:**

- Regarding the summary:
Our position is *not* about NeSy AI, nor are our pillars solely based on the ARC-AGI benchmark, indicating that this reviewer did neither completely read nor understand the paper.

- Clarity of arguments and definitions.
Multiple prerequisite arguments and definitions are from Chollet 2019 (as referenced in the paper multiple times. For a researcher/reviewer proficient in this area of research, some 5 years of background knowledge must be expected. Our paper shall only provide novel ideas.

- Regarding writing quality:
We doubt the representativeness of the reviewer's perception here, as all other researchers who read the manuscript had no such issues.

- Regarding perceived vagueness:
See discussion in Appendix H.1 lines 1010-1023. Additionally, note that we dedicated 3.5 pages describing all components (see Chapter " 3 Pillars of Efficient Generalization"). The perspective we propose is a very general one (meta-level). Some details cannot be specified without ending up in concrete methodological instantiations. For examples of such (and their diversity), see the references/sources we provide. We try to be as concrete as possible without loss of general correctness.

- Regarding the reviewer's questions:
First, we would hope to see more foundational research on the pillars' synergy effects, enabling generalization. Afterwards, applied research would follow with more comprehensive architectures. Currently, we see the trend of overoptimizing one pillar (e.g. knowledge acquisition) and adding missing components later (e.g. reasoning structures, see LRMs). In the long run, this trend must switch to be effective: generalization/breadth first, engineering/depth later.

- Regarding all other mentioned aspects:
We strongly urge the reviewer to carefully re-read our paper and the first few paragraphs of the NeurIPS 2025 Call for Position Papers ( https://neurips.cc/Conferences/2025/CallForPositionPapers ) and reconsider the evaluation of our paper.

---

### Meta-Review · Area_Chair_q6ur · 2025-09-17

**Rating:** 3
**Confidence:** 3

**Strengths:**

The paper is clearly motivated and framing a timely problem. It provides a structured six-pillar framework that highlights key components (although some reviewers found it otherwise). It could contribute to the broader audience on building more generalizable AI systems.

**Weaknesses:**

The main weaknesses of the paper is its lack of clarity and rigorous. Reviewers pointed out that several terms and proposed pillars are vague or insufficiently defined, making the overall argument hard to follow (unless as the author argued, substantially refer to previous literature). The pillars themselves are not convincingly shown to be either necessary or uniquely important. Additionally, the scope of generalization is not clearly specified.

**Questions:**

While the author has argued that "the six pillars are the minimal amount of complexity required to achieve efficient generalization", are there any relative order of importance? In other words, if we had to prioritize among the six proposed pillars, which one or two do you believe are most essential for achieving efficient AI generalization in practice?

**Thoroughness:**

4

---

### Decision · Program_Chairs · 2025-09-26

Reject